# ON ACCELERATED PERCEPTRONS AND BEYOND

**Guanghui Wang**[1]**, Rafael Hanashiro**[2]**, Etash Guha**[1]**, Jacob Abernethy**[1,3]

[1]College of Computing, Georgia Tech, Atlanta, GA, USA
[2]Department of Electrical Engineering and Computer Science, MIT, Cambridge, MA, USA
[3]Google Research, Atlanta, GA 30309
{gwang369,etash}@gatech.edu, rafah@mit.edu, abernethyj@google.com

## ABSTRACT

The classical Perceptron algorithm of Rosenblatt can be used to find a linear threshold function to correctly classify $n$ linearly separable data points, assuming the classes are separated by some margin $\gamma > 0$. A foundational result is that Perceptron converges after $O(1/\gamma^2)$ iterations. There have been several recent works that managed to improve this rate by a quadratic factor, to $O(\sqrt{\log n}/\gamma)$, with more sophisticated algorithms. In this paper, we unify these existing results under one framework by showing that they can all be described through the lens of solving min-max problems using modern acceleration techniques, mainly through *optimistic* online learning. We then show that the proposed framework also leads to improved results for a series of problems beyond the standard Perceptron setting. Specifically, a) for the *margin maximization* problem, we improve the state-of-the-art result from $O(\log t/t^2)$ to $O(1/t^2)$, where $t$ is the number of iterations; b) we provide the first result on identifying the *implicit bias* property of the classical Nesterov's accelerated gradient descent (NAG) algorithm, and show NAG can maximize the margin with an $O(1/t^2)$ rate; c) for the classical *p-norm Perceptron* problem, we provide an algorithm with $O(\sqrt{(p-1)\log n}/\gamma)$ convergence rate, while existing algorithms suffer the $O((p-1)/\gamma^2)$ convergence rate.

## 1 INTRODUCTION

In this paper, we revisit the problem of learning a linear classifier, which is one of the most important and fundamental tasks of machine learning (Bishop, 2007). In this problem, we are given a set $\mathcal{S}$ of $n$ training examples, and the goal is to find a linear classifier that correctly separates $\mathcal{S}$ as fast as possible. The most well-known algorithm is Perceptron (Rosenblatt, 1958), which can converge to a perfect (mistake-free) classifier after $\Omega(1/\gamma^2)$ number of iterations, provided the data is linearly separable with some margin $\gamma > 0$ (Novikoff, 1962). Over subsequent decades, many variants of Perceptron have been developed (Aizerman, 1964; Littlestone, 1988; Wendemuth, 1995; Freund & Schapire, 1999; Cesa-Bianchi et al., 2005, to name a few). However, somewhat surprisingly, there has been little progress in substantially improving the fundamental Perceptron iteration bound presented by Novikoff (1962). It is only recently that a number of researchers have discovered *accelerated* variants of the Perceptron with a faster $\Omega(\sqrt{\log n}/\gamma)$ iteration complexity, although with a slower per-iteration cost. These works model the problem in different ways, e.g., as a non-smooth optimization problem or an empirical risk minimization task, and they have established faster rates using sophisticated optimization tools. Soheili & Pena (2012) put forward the *smooth Perceptron*, framing the objective as a non-smooth strongly-concave maximization and then applying *Nestrov's excessive gap technique* (NEG, Nesterov, 2005). Yu et al. (2014) proposed the *accelerated Perceptron* by furnishing a convex-concave objective that can be solved via the mirror-prox method (Nemirovski, 2004). Ji et al. (2021) put forward a third interpretation, obtaining the accelerated rate by minimizing the empirical risk under exponential loss with a momentum-based normalized gradient descent algorithm.

Following this line of research, in this paper, we present a *unified* analysis framework that reveals the exact relationship among these methods that share the same order of convergence rate. Moreover, we show that the proposed framework also leads to improved results for various problems beyond the standard Perceptron setting. Specifically, we consider a general zero-sum game that involves

two players (Abernethy et al., 2018): a main player that chooses the classifier, and an auxiliary player that picks a distribution over data. The two players compete with each other by performing *no-regret online learning* algorithms (Hazan, 2016; Orabona, 2019), and the goal is to find the equilibrium of some convex-concave function. We show that, under this dynamic, all of the existing accelerated Perceptrons can find their equivalent forms. In particular, these perceptrons can be described as a dynamic where two players solving the game via performing *optimistic online learning* strategies (Rakhlin & Sridharan, 2013), which is one of the most important classes of algorithms in online learning. Note that implementing online learning algorithms (even optimistic strategies) to solve zero-sum games has already been extensively explored (e.g., Rakhlin & Sridharan, 2013; Daskalakis et al., 2018; Wang & Abernethy, 2018; Daskalakis & Panageas, 2019). However, we emphasize that our main novelty lies in showing that all of the existing accelerated Perceptrons, developed with advanced algorithms from different areas, can be perfectly described under this unified framework. It greatly simplifies the analysis of accelerated Perceptrons, as their convergence rates can now be easily obtained by plugging-in off-the-shelf regret bounds of optimistic online learning algorithms. Moreover, the unified framework reveals a close connection between the smooth Perceptron and the accelerated Perceptron of Ji et al. (2021):

**Theorem 1** (informal). *Smooth Perceptron and the accelerated Perceptron of Ji et al. (2021) can be described as a dynamic where the two players employ the optimistic-follow-the-regularized-leader (OFTRL) algorithm to play. The main difference is that the smooth Perceptron outputs the* weighted average *of the main player's historical decisions, while the accelerated Perceptron of Ji et al. (2021) outputs the* weighted sum.

Beyond providing a deeper understanding of accelerated Perceptrons, our framework also provides improved new results for several other important areas:

- *Implicit bias* analysis. The seminal work of Soudry et al. (2018) shows that, for linearly separable data, minimizing the empirical risk with the vanilla gradient descent (GD) gives a classifier which not only has zero training error (thus can be used for linear separation), but also *maximizes the margin*. This phenomenon characterizes the *implicit bias* of GD, as it implicitly prefers the ($\ell_2$-)maximal margin classifier among all classifiers with a positive margin, and analysing the implicit bias has become an important tool for understanding why *classical* optimization methods generalize well for supervised machine learning problems. Soudry et al. (2018) show that GD can maximize the margin in an $O(1/\log t)$ rate, and this is later improved to $O(1/\sqrt{t})$ (Nacson et al., 2019) and then $O(1/t)$ (Ji & Telgarsky, 2021). The state-of-the-art algorithm is proposed by Ji et al. (2021), who show that their proposed momentum-based GD has an $O(\log t/t^2)$ margin-maximization rate. In this paper, we make two contributions toward this direction:

  1. We show that, under our analysis framework, we can easily improve the margin maximization rate of the algorithm of Ji et al. (2021) from $O(\log t/t^2)$ to $O(1/t^2)$;
  2. Although previous work has analyzed the implicit bias of GD and momentum-based GD, it is still unclear how the classical Nesterov's accelerated gradient descent (NAG, Nesterov, 1988) will affect the implicit bias. In this paper, through our framework, we show that NAG with appropriately chosen parameters also enjoys an $O(1/t^2)$ margin-maximization rate. To our knowledge, it is the first time the implicit bias property of NAG is proved.

- $p$-norm Perceptron. Traditional work on Perceptrons typically assumes the feature vectors lie in an $\ell_2$-ball. A more generalized setting is considered in Gentile (2000), who assumes the feature vectors lie inside an $\ell_p$-ball, with $p \in [2, \infty)$. Their proposed algorithm requires $O(p/\gamma^2)$ number of iterations to find a zero-error classifier. In this paper, we develop a new Perceptron algorithm for this problem under our framework based on optimistic strategies, showing that it enjoys an accelerated $O(\sqrt{p \log n}/\gamma)$ rate.

## 2 RELATED WORK

This section briefly reviews the related work on Perceptron algorithms, implicit-bias analysis, and game theory. The background knowledge on (optimistic) online learning is presented in the Preliminaries (Section 3).

**Accelerated Perceptrons and $p$-norm Perceptron** The study of Perceptron algorithms has an extensive history dating back to the mid-twentieth century (Rosenblatt, 1958; Novikoff, 1962). However, it is only recently that progress on improving the fundamental $\Omega(1/\gamma^2)$ iteration bound of the vanilla Perceptron in the standard setting has been made. Specifically, the smooth Perceptron proposed by Soheili & Pena (2012) achieves an $\Omega(\sqrt{\log n}/\gamma)$ rate by maximizing an auxiliary non-smooth strongly-concave function with NEG (Nesterov, 2005). The same accelerated rate is later obtained by two other work (Yu et al., 2014; Ji et al., 2021). The former considers a bi-linear saddle point problem and employs the mirror-prox method (Nemirovski, 2004). The latter applies momentum-based GD to minimize the empirical risk with exponential loss. In this paper, we show that these algorithms can be unitedly analysed under our framework.

Apart from the above accelerated Perceptrons, there exists another class of algorithms which enjoy an $O(\text{poly}(n)\log(1/\gamma))$ convergence rate (Dunagan & Vempala, 2004; Peña & Soheili, 2016; Dadush et al., 2020). These methods typically call (accelerated) Perceptrons as a subroutine, and then apply the re-scaling technique to adjust the decision periodically. Although the dependence on $\gamma$ becomes better, the polynomial dependence on $n$ of these methods makes them computationally inefficient for large-scale data sets. In this paper, we focus on accelerated Perceptrons with $O(\sqrt{\log n}/\gamma)$ rate, and leave explaining the re-scaling type algorithms as future work.

The $p$-norm Perceptron problem (Gentile & Littlestone, 1999) is a natural extension of the classical Perceptron setting, which assumes the $p$-norm of the feature vectors are bounded, where $2 \le p < \infty$. Gentile (2001) shows that a mirror-descent-style update guarantees an $O(p/\gamma^2)$ convergence rate. By contrast, our proposed algorithm achieves a tighter $O(\sqrt{p\log n}/\gamma)$ convergence rate.

**Implicit-bias analysis** In many real-world applications, directly minimizing the empirical risk (without any regularization) by first-order methods can provide a model which, not only enjoys low training error, but also generalizes well (Soudry et al., 2018). This is usually considered as the *implicit bias introduced by the optimization methods* (Soudry et al., 2018). Explaining this phenomenon is a crucial step towards understanding the generalization ability of *commonly-used* optimization methods. For linear separable data, Soudry et al. (2018) proves that, when minimizing the empirical risk with exponential loss, the vanilla GD can maximize the margin in an $O(1/\log t)$ rate. This result implies the implicit bias of GD towards the $\ell_2$-maximal margin classifier. Later, Nacson et al. (2019) show that GD with a function-value-dependant decreasing step-size enjoys an $O(1/\sqrt{t})$ margin-maximization rate. Ji & Telgarsky (2021) improve this result to $O(1/t)$ by employing a faster-decreasing step size. Ji et al. (2021) design a momentum-based GD that maximizes the margin with an $O(\log t/t^2)$ rate. However, it remained unclear whether this rate could be further improved and how to analyze the margin maximization ability of other classical optimization methods such as NAG. In this paper, we provide positive answers to both questions. Fianlly, we note that (Ramdas & Pena, 2016) prove a varint of the Perceptron algorithm named normalized perceptron can also converge to a maximum margin classifer in an $O(1/\sqrt{t})$ convergence rate.

**Games and no-regret dynamics** Our framework is motivated by the line of research that links optimization methods for convex optimization to equilibrium computation with no-regret dynamics. The seminal work of Rakhlin & Sridharan (2013) recovers the classical mirror-prox method (Nemirovski, 2004) with Optimistic online mirror descent. Abernethy & Wang (2017) show that the well-known Frank-Wolfe algorithm can be seen as applying (optimistic) online algorithms to compute the equilibrium of a special zero-sum game called *Fenchel game*, which is constructed via the Fenchel duality of the objective function. Later, researchers demonstrate that other classical optimization methods for smooth optimization, such as Heavy-ball and NAG, can also be described similarly (Abernethy et al., 2018; Wang & Abernethy, 2018; Wang et al., 2021). We highlight the differences between this work and the previous ones: 1) Our analysis does not involve the Fenchel game or Fenchel duality; instead, we directly work on the (regularized) min-max game designed for linear classification. 2) Most of the previous work mainly focus on understanding optimization algorithms for *smooth* optimization, and it was unclear how to understand algorithm such as NEG under the game framework. 3) Although both Abernethy et al. (2018) and our work analyze NAG, the goals are significantly different: Abernethy et al. (2018) focus on the optimization problem itself, while we consider how minimizing the empirical risk would affect the *margin*. 4) To our knowledge, the link between the implicit-bias problems and no-regret dynamics is also new.

## 3 PRELIMINARIES

**Notation.** We use lower case bold face letters $\mathbf{x}, \mathbf{y}$ to denote vectors, lower case letters $a, b$ to denote scalars, and upper case letters $A, B$ to denote matrices. For a vector $\mathbf{x} \in \mathbb{R}^d$, we use $x_i$ to denote the $i$-th component of $\mathbf{x}$. For a matrix $A \in \mathbb{R}^{n \times d}$, let $A_{(i,:)}$ be its $i$-th row, $A_{(:,j)}$ the $j$-th column, and $A_{(i,j)}$ the $i$-th element of the $j$-th column. We use $\| \cdot \|$ to denote a general norm, $\| \cdot \|_*$ its dual norm, and $\| \cdot \|_p$ the $\ell_p$-norm. For a positive integer $n$, we denote the set $\{1, \dots, n\}$ as $[n]$, and the $n$-dimensional simplex as $\Delta^n$. Let $E : \Delta^n \mapsto \mathbb{R}$ be the negative entropy function, defined as $E(\mathbf{p}) = \sum_{i=1}^{n} p_i \log p_i, \forall \mathbf{p} \in \Delta^n$. For some strongly convex function $R(\mathbf{w}) : \mathcal{W} \mapsto \mathbb{R}$, define the Bregman divergence between any two points $\mathbf{w}, \mathbf{w}' \in \mathcal{W}$ as:

$$D_R(\mathbf{w}, \mathbf{w}') = R(\mathbf{w}) - R(\mathbf{w}') - (\mathbf{w} - \mathbf{w}')^\top \nabla R(\mathbf{w}').$$

**Online convex optimization (OCO)** Here we review a general *weighted* OCO framework, proposed by Abernethy et al. (2018). In each round $t = 1, \dots, T$ of this paradigm, a learner first chooses a decision $\mathbf{z}_t$ from a convex set $\mathcal{Z} \subseteq \mathbb{R}^d$, then observes a loss function $f_t(\cdot) : \mathcal{Z} \to \mathbb{R}$ as well as a weight $\alpha_t > 0$, and finally updates the decision. The performance of the learner is measured by the *weighted* regret, which is defined as $R_T = \sum_{t=1}^{T} \alpha_t f_t(\mathbf{z}_t) - \min_{\mathbf{z} \in \mathcal{Z}} \sum_{t=1}^{T} \alpha_t f_t(\mathbf{z})$.

Perhaps the most natural method for OCO is follow-the-leader (FTL), which simply picks the empirically best decision at each round: $\mathbf{z}_t = \operatorname{argmin}_{\mathbf{z} \in \mathcal{Z}} \sum_{i=1}^{t-1} \alpha_i f_i(\mathbf{z})$. However, FTL is unstable, and one can easily find counter-examples where FTL suffers linear regret (Shalev-Shwartz, 2011). A classical way to address this limitation is by adding a regularizer to the objective: $\mathbf{z}_t = \operatorname{argmin}_{\mathbf{z} \in \mathcal{Z}} \eta \sum_{i=1}^{t-1} \alpha_i f_i(\mathbf{z}) + D_R(\mathbf{z}, \mathbf{z}_0)$, where $\eta > 0$ is the step size, and $\mathbf{z}_0$ is the initial decision. This method is called follow-the-regularized-leader (Hazan, 2016), and it can achieve a sub-linear regret bound with appropriately chosen $\eta$. Moreover, tighter bounds are also possible in favored cases with more advanced techniques. In this paper, we consider FTRL equipped with optimistic strategies (i.e., Optimistic FTRL, Rakhlin & Sridharan, 2013; Orabona, 2019), given by

$$\text{OFTRL}[R, \mathbf{z}_0, \psi_t, \eta, \mathcal{Z}] : \mathbf{z}_t = \operatorname{argmin}_{\mathbf{z} \in \mathcal{Z}} \eta \left[ \sum_{i=1}^{t-1} \alpha_i f_i(\mathbf{z}) + \alpha_t \psi_t(\mathbf{z}) \right] + D_R(\mathbf{z}, \mathbf{z}_0),$$

where an additional function $\psi_t$ is added, which is an approximation of the next loss $f_t$. A tighter regret bound can be achieved when $\psi_t$ is close enough to $f_t$. Next, we introduce two special cases of OFTRL:

$$\text{OFTL}[\psi_t, \mathcal{Z}] : \mathbf{z}_t = \operatorname*{argmin}_{\mathbf{z} \in \mathcal{Z}} \sum_{j=1}^{t-1} \alpha_j f_j(\mathbf{z}) + \alpha_t \psi_t(\mathbf{z})$$

$$\text{FTRL}^+[R, \mathbf{z}_0, \eta, \mathcal{Z}] : \mathbf{z}_t = \operatorname*{argmin}_{\mathbf{z} \in \mathcal{Z}} \eta \cdot \sum_{j=1}^{t} \alpha_j f_j(\mathbf{z}) + \mathcal{D}_R(\mathbf{z}, \mathbf{z}_0).$$

The first one is Optimistic FTL, where the regularizer is set to be zero. The second algorithm is FTRL$^+$, which uses $f_t$ as the optimistic function $\psi_t$. Finally, we note that, apart from FTL-type algorithms, there also exists optimistic methods that are developed based on mirror descent, such as optimistic online mirror decent (OMD, Rakhlin & Sridharan, 2013). Due to page limitation, the details of OMD and the regret bounds of these OCO algorithms are postponed to the Appendix A.

**No-regret dynamics for zero-sum game** Finally, we introduce the framework for using no-regret online algorithms to solve a zero-sum game. Consider the following general two-player game:

$$\max_{\mathbf{w} \in \mathcal{W}} \min_{\mathbf{p} \in \mathcal{Q}} g(\mathbf{w}, \mathbf{p}), \tag{1}$$

where $g(\mathbf{w}, \mathbf{p})$ is a concave-convex function, and $\mathcal{W}$ and $\mathcal{Q}$ are convex sets. The no-regret framework for solving (1) is presented in Protocol 1. In each round $t$ of this procedure, the $\mathbf{w}$-player first picks a decision $\mathbf{w}_t \in \mathcal{W}$. Then, the $\mathbf{p}$-player observes its loss $\ell_t(\cdot) = g(\mathbf{w}_t, \cdot)$ as well as a weight $\alpha_t$. After that, the $\mathbf{p}$-player picks the decision $\mathbf{p}_t$, and passes it to the $\mathbf{w}$-player. As a consequence, the $\mathbf{w}$-player observes its loss $h_t(\cdot) = -g(\cdot, \mathbf{p}_t)$ and weight $\alpha_t$. Note that both $\ell_t$ and $h_t$ are convex. Denote the weighted regret bounds of the two players as $R^{\mathbf{w}}$ and $R^{\mathbf{p}}$, respectively. Then, we have the following classical conclusion. The proof is shown in Appendix B.

**Theorem 2.** *Define* $m(\mathbf{w}) = \min_{\mathbf{p} \in \mathcal{Q}} g(\mathbf{w}, \mathbf{p})$*, and* $\overline{\mathbf{w}}$ *the weighted average of* $\{\mathbf{w}_t\}_{t=1}^{T}$*. Then we have that* $\forall \mathbf{w} \in \mathcal{W}, m(\mathbf{w}) - m(\overline{\mathbf{w}}_T) \leq \left( \sum_{t=1}^{T} \alpha_t \right)^{-1} (R^{\mathbf{p}} + R^{\mathbf{w}})$.

---

**Protocol 1** No-regret dynamics with weighted OCO

---

1: **Input**: $\{\alpha_t\}_{t=1}^T, \mathbf{w}_0, \mathbf{p}_0$.
2: **Input**: $\mathsf{OL}^{\mathbf{w}}, \mathsf{OL}^{\mathbf{p}}$. // The online algorithms for choosing $\mathbf{w}$ and $\mathbf{p}$.
3: **for** $t = 1, \ldots, T$ **do**
4: $\quad \mathbf{w}_t \leftarrow \mathsf{OL}^{\mathbf{w}}$;
5: $\quad \mathsf{OL}^{\mathbf{p}} \leftarrow \alpha_t, \ell_t(\cdot)$; // Define $\ell_t(\cdot) = g(\mathbf{w}_t, \cdot)$
6: $\quad \mathbf{p}_t \leftarrow \mathsf{OL}^{\mathbf{p}}$;
7: $\quad \mathsf{OL}^{\mathbf{w}} \leftarrow \alpha_t, h_t(\cdot)$; // Define $h_t(\cdot) = -g(\cdot, \mathbf{p}_t)$
8: **end for**
9: **Output**: $\overline{\mathbf{w}}_T = \frac{1}{\sum_{t=1}^T \alpha_t} \sum_{t=1}^T \alpha_t \mathbf{w}_t$.

---

# 4 UNDERSTANDING EXISTING ACCELERATED PERCEPTRONS

In this section, we first introduce our main topic, i.e., the binary linear classification problem, and then present the three accelerated Perceptrons and their equivalent forms under Protocol 1. For clarity, we use $\mathbf{v}_t$ to denote the classifier updates in the original algorithms, and $\mathbf{w}_t$ the corresponding updates in the equivalent forms under Protocol 1.

## 4.1 BINARY LINEAR CLASSIFICATION AND PERCEPTRON

We focus on the binary linear classification problem, which dates back to the pioneering work of Rosenblatt (1958). Let $\mathcal{S} = \{\mathbf{x}^{(i)}, y^{(i)}\}_{i=1}^n$ be a linear-separable set of $n$ training examples, where $\mathbf{x}^{(i)} \in \mathbb{R}^d$ is the feature vector of the $i$-th example, and $y^{(i)} \in \{-1, 1\}$ is the corresponding label. The goal is to efficiently find a linear classifier $\mathbf{w} \in \mathbb{R}^d$ that correctly separates all data points. More formally, let $A = [y^{(1)}\mathbf{x}^{(1)}, \ldots, y^{(n)}\mathbf{x}^{(n)}]^\top$ be the matrix that contains all of the data. Then we would like to find a $\mathbf{w} \in \mathbb{R}^d$ such that

$$\min_{i \in [n]} A_{(i,:)}\mathbf{w} > 0. \tag{2}$$

This goal can be reformulated as a min-max optimization problem:

$$\max_{\mathbf{w} \in \mathbb{R}^d} \min_{i \in [n]} A_{(i,:)}\mathbf{w} = \max_{\mathbf{w} \in \mathbb{R}^d} \min_{\mathbf{p} \in \Delta^n} \mathbf{p}^\top A\mathbf{w}, \tag{3}$$

where, at the RHS of (3), $\min_{i \in [n]} A_{(i,:)}\mathbf{w}$ is rewritten as $\min_{\mathbf{p} \in \Delta^n} \mathbf{p}^\top A\mathbf{w}$. The two expressions are equivalent since the optimal distribution $\mathbf{p} \in \Delta^n$ will always put all weight on *one* training example (i.e., one row) in $A$. For any $\mathbf{w} \in \mathbb{R}^d$, let $\gamma(\mathbf{w}) = \min_{\mathbf{p} \in \Delta^n} \mathbf{p}^\top A\mathbf{w}$ be the *margin* of $\mathbf{w}$, and we introduce the following standard assumption.

**Assumption 1.** *We assume that feature vectors are bounded, i.e., $\|\mathbf{x}^{(i)}\|_2 \leq 1$ for all $i \in [n]$, and that there exists a $\mathbf{w}^* \in \mathbb{R}^d$, such that $\|\mathbf{w}^*\|_2 = 1$ and $\gamma(\mathbf{w}^*) = \gamma > 0$.*

## 4.2 SMOOTH PERCEPTRON

In order to solve (3), the vanilla Perceptron repeatedly moves the direction of the classifier to that of examples on which it performs badly. However, this greedy policy only yields a sub-optimal convergence rate. To address this problem, Soheili & Pena (2012) propose the Smooth Perceptron, and the pseudo-code is summarized in the first box in Algorithm 1. The key idea is to find a classifier $\mathbf{v} \in \mathbb{R}^d$ that maximizes the following $\ell_2$-regularized function:

$$\psi(\mathbf{v}) = -\frac{1}{2}\|\mathbf{v}\|_2^2 + \min_{\mathbf{q} \in \Delta^n} \mathbf{q}^\top A\mathbf{v}, \tag{4}$$

which is *non-smooth strongly-concave* with respect to $\mathbf{v}$. Under Assumption 1, Soheili & Pena (2012) show that the maximum value of $\psi(\mathbf{v})$ is $\max_{\mathbf{v} \in \mathbb{R}^d} \psi(\mathbf{v}) = \gamma^2/2$, and a classifier $\mathbf{v} \in \mathbb{R}^d$ satisfies (2) when $\psi(\mathbf{v}) \geq 0$. In order to solve (4), Soheili & Pena (2012) apply the classical Nesterov's excessive gap technique (Nesterov, 2005) for strongly concave functions. This algorithm introduces a smoothed approximation of (4), which is parameterized by some constant $\mu > 0$, and defined as

$$\psi_\mu(\mathbf{v}) = -\frac{1}{2}\|\mathbf{v}\|_2^2 + \min_{\mathbf{q} \in \Delta^n} \left[\mathbf{q}^\top A\mathbf{v} + \mu D_E\left(\mathbf{q}, \frac{1}{n}\right)\right], \tag{5}$$

---

**Algorithm 1** Smooth Perceptron (Soheili & Pena, 2012)

---

**Initialization :** $\theta_0 = \frac{2}{3}$, $\mu_0 = 4$, $\mathbf{v}_0 = \frac{A^\top \mathbf{1}}{n}$
**Initialization :** $\mathbf{q}_0 = \mathbf{q}_{\mu_0}(\mathbf{v}_0)$, where $\mathbf{q}_\mu(\mathbf{v}) = \operatorname{argmin}_{\mathbf{q} \in \Delta^n} \mathbf{q}^\top A \mathbf{v} + \mu D_E\left(\mathbf{q}, \frac{1}{n}\right)$.
**for** $t = 1, \ldots, T - 1$ **do**
$\quad \mathbf{v}_t = (1 - \theta_{t-1})(\mathbf{v}_{t-1} + \theta_{t-1} A \mathbf{q}_{t-1}) + \theta_{t-1}^2 A \mathbf{q}_{\mu_{t-1}}(\mathbf{v}_{t-1})$
$\quad \mu_t = (1 - \theta_{t-1})\mu_{t-1}$
$\quad \mathbf{q}_t = (1 - \theta_{t-1})\mathbf{q}_{t-1} + \theta_{t-1}\mathbf{q}_{\mu_t}(\mathbf{v}_t)$
$\quad \theta_t = \frac{2}{t+3}$
**end for**
**Output:** $\mathbf{v}_{T-1}$

---

$$\mathsf{OL}^\mathbf{w} = \mathrm{OFTL}\left[h_{t-1}(\cdot), \mathbb{R}^d\right] \Leftrightarrow \mathbf{w}_t = \operatorname*{argmin}_{\mathbf{w} \in \mathbb{R}^d} \sum_{j=1}^{t-1} \alpha_j h_j(\mathbf{w}) + \alpha_t h_{t-1}(\mathbf{w})$$

$$\mathsf{OL}^\mathbf{p} = \mathrm{FTRL}^+\left[E(\cdot), \frac{\mathbf{1}}{n}, \frac{1}{4}, \Delta^n\right] \Leftrightarrow \mathbf{p}_t = \operatorname*{argmin}_{\mathbf{p} \in \Delta^n} \frac{1}{4} \sum_{s=1}^{t} \alpha_s \ell_s(\mathbf{p}) + \mathcal{D}_E\left(\mathbf{p}, \frac{\mathbf{1}}{n}\right)$$

**Output:** $\overline{\mathbf{w}}_T$

---

which bounds the function in (4) from below, i.e., $\forall \mathbf{v} \in \mathbb{R}^d, \mu > 0, \psi_\mu(\mathbf{v}) \leq \psi(\mathbf{v}) + \mu \log n$. Then, the algorithm performs a sophisticated update rule such that the excessive gap condition holds $\forall t \geq 1$: $\gamma^2/2 \leq \|A\mathbf{v}_t\|^2/2 \leq \psi_\mu(\mathbf{v}_t)$. We refer to Soheili & Pena (2012) for more details.

Soheili & Pena (2012) show that the smooth Perceptron can output a positive-margin classifier after $\Omega(\sqrt{\log n}/\gamma)$ iterations. However, the analysis is quite involved, which heavily relies on the complicated relationship between $\psi(\mathbf{v}_t)$, $\psi_\mu(\mathbf{v}_t)$ and $\|A\mathbf{v}_t\|^2$. In the following, we provide a no-regret explanation of this algorithm and then show that the convergence rate can be easily obtained under our framework. Specifically, we define the objective function in Protocol 1 as

$$g(\mathbf{w}, \mathbf{p}) = \mathbf{p}^\top A \mathbf{w} - \frac{1}{2}\|\mathbf{w}\|_2^2, \tag{6}$$

and provide the equivalent expression in the second box of Algorithm 1. More specifically, we have the following proposition. The proof is given in Appendix C.1.

**Proposition 1.** *Let $\alpha_t = t$. Then the two interpretations of the smooth Perceptron in Algorithm 1 are the same, in the sense that $\mathbf{v}_{T-1} = \overline{\mathbf{w}}_T$, and $\mathbf{q}_{T-1} = \frac{\sum_{t=1}^T \alpha_t \mathbf{p}_t}{\sum_{t=1}^T \alpha_t}$.*

Proposition 1 shows that, under Protocol 1, the smooth Perceptron can be seen as optimizing (6) by implementing two online learning algorithms: in each round $t$, the $\mathbf{w}$-player applies OFTL with $\psi_t = h_{t-1}$, a decision which the $\mathbf{p}$-player subsequently observes and responds with FTRL$^+$. Moreover, we obtain theoretical guarantees for the smooth Perceptron based on Theorem 2, matching the convergence rate provided by Soheili & Pena (2012).

**Theorem 3.** *Let $\alpha_t = t$. Under Protocol 1, the regret of the two players of Algorithm 1 is bounded by $R^\mathbf{w} \leq 2\sum_{t=1}^T \|\mathbf{p}_t - \mathbf{p}_{t-1}\|_1^2$ and $R^\mathbf{p} \leq 4\log n - 2\sum_{t=1}^T \|\mathbf{p}_t - \mathbf{p}_{t-1}\|_1^2$. Moreover, $\overline{\mathbf{w}}_T$ has non-negative margin when $T = \Omega(\frac{\sqrt{\log n}}{\gamma})$.*

## 4.3 ACCELERATED PERCEPTRON VIA ERM

Since binary linear classification is a (and perhaps the most fundamental) supervised machine learning problem, it can be naturally solved via empirical risk minimization (ERM). This idea is adopted by Ji et al. (2021), who consider the following ERM problem:

$$\min_{\mathbf{v} \in \mathbb{R}^d} R(\mathbf{v}) = \frac{1}{n}\sum_{i=1}^n \ell(-y^{(i)}\mathbf{v}^\top \mathbf{x}^{(i)}), \tag{7}$$

where $\ell(z) = \exp(z)$ is the exponential loss. To minimize (7), Ji et al. (2021) propose the a normalized momentum-based gradient descent (NMGD) algorithm, and show that NMGD can converge to

---

**Algorithm 2** Accelerated Perceptron of Ji et al. (2021)

---

**Input**: $\mathbf{q}_0 = \frac{1}{n}, \mathbf{v}_0 = \mathbf{0}, \mathbf{g}_0 = \mathbf{0}$.
**for** $t = 1, \ldots, T$ **do**
    Set $\theta_{t-1} = \frac{t}{2(t+1)}, \beta_t = \frac{t}{t+1}$
    $\mathbf{v}_t = \mathbf{v}_{t-1} - \theta_{t-1}(\mathbf{g}_{t-1} - A^\top \mathbf{q}_{t-1})$
    **for** $i = 1, \ldots, n$ **do**
        $q_{t,i} = \frac{\exp(-y^{(i)} \mathbf{v}_t^\top \mathbf{x}^{(i)})}{\sum_{j=1}^n \exp(-y^{(j)} \mathbf{v}_t^\top \mathbf{x}^{(j)})}$
    **end for**
    $\mathbf{g}_t = \beta_t(\mathbf{g}_{t-1} - A^\top \mathbf{q}_t)$
**end for**
**Output**: $\mathbf{v}_T$

---

$$\mathsf{OL}^{\mathbf{w}} = \mathsf{OFTL}\left[h_{t-1}(\cdot), \mathbb{R}^d\right] \Leftrightarrow \mathbf{w}_t = \underset{\mathbf{w} \in \mathbb{R}^d}{\operatorname{argmin}} \sum_{j=1}^{t-1} \alpha_j h_j(\mathbf{w}) + \alpha_t h_{t-1}(\mathbf{w})$$

$$\mathsf{OL}^{\mathbf{p}} = \mathsf{FTRL}^+\left[E(\cdot), \frac{1}{n}, \frac{1}{4}, \Delta^n\right] \Leftrightarrow \mathbf{p}_t = \underset{\mathbf{p} \in \Delta^n}{\operatorname{argmin}} \frac{1}{4} \sum_{s=1}^{t} \alpha_s \ell_s(\mathbf{p}) + \mathcal{D}_E\left(\mathbf{p}, \frac{\mathbf{1}}{n}\right)$$

**Output**: $\overline{\mathbf{w}}_T$

---

a classifier with a positive margin after $O(1/\gamma^2)$ iterations. Based on the property of the normalized gradient, they also provide primal-dual form of the algorithm (presented in the first-box of Algorithm 2), which can be considered as applying Nesterov's accelerated gradient descent in the *dual space* (Ji et al., 2021) .

For the accelerated Perceptron of Ji et al. (2021), we set $g(\mathbf{w}, \mathbf{p}) = \mathbf{p}^\top A\mathbf{w} - \frac{1}{2}\|\mathbf{w}\|_2^2$ and provide its equivalent form at the second box in Algorithm 2. Specifically, we have the following proposition.

**Proposition 2.** *Let $\alpha_t = t$. Then the two interpretations of the accelerated Perceptron in Algorithm 2 are the same, in the sense that $\mathbf{q}_T = \mathbf{p}_T$ and $\mathbf{v}_T = \frac{1}{4}\sum_{t=1}^T \alpha_t \mathbf{w}_t = \frac{1}{4}\left(\sum_{t=1}^T \alpha_t\right) \cdot \overline{\mathbf{w}}_T$.*

**Remark** Note that in Ji et al. (2021), the parameter $\theta_{t-1}$ is set to be 1. We observe that this causes a mismatch between the weights and $\mathbf{w}_t'$s in the output (when $\theta_{t-1} = 1$, $\mathbf{v}_T$ will become $\sum_{t=1}^T \alpha_{t+1} \mathbf{w}_t$ instead of $\sum_{t=1}^T \alpha_t \mathbf{w}_t$). Our no-regret analysis suggests that $\theta_{t-1}$ should be configured as $\frac{t}{2(t+1)}$, and later we will show that this procedure helps eliminate the $\log t$ factor in the convergence rate.

The proposition above reveals that the smooth Perceptron and the accelerated Perceptron of Ji et al. (2021) are closely related. The main difference is that the accelerated Perceptron of Ji et al. (2021) outputs the weighted sum of all $\mathbf{w}_t$'s, instead of the weighted average. The rationale behind this phenomenon is that, instead of the margin, Ji et al. (2021) use the *normalized* margin to measure the performance of the algorithm, defined as $\overline{\gamma}(\mathbf{v}) = \min_{\mathbf{p} \in \Delta^n} \mathbf{p}^\top A\mathbf{v}/\|\mathbf{v}\|_2$. They not only prove $\overline{\gamma}(\mathbf{v}_T) > 0$ (which directly implies $\gamma(\mathbf{v}_T) > 0$), but also show $\overline{\gamma}(\mathbf{v}_T) = \Omega(\gamma - \frac{\log T}{\gamma T^2})$. This guarantee is more powerful than the previous results, as it implies that the *normalized margin can be maximized* in an $O(\log t/t^2)$ rate. When $t$ approaches $\infty$, the direction of $\mathbf{v}_T$ will converge to that of the maximal margin classifier. We show that a better margin-maximization rate can be directly obtained under our framework with the parameter setting in Algorithm 2.

**Theorem 4.** *Let $\alpha_t = t$. Under Protocol 1, the regret of the two players of Algorithm 2 is bounded by $R^{\mathbf{w}} \leq 2\sum_{t=1}^T \|\mathbf{p}_t - \mathbf{p}_{t-1}\|_1^2$ and $R^{\mathbf{p}} \leq 4\log n - 2\sum_{t=1}^T \|\mathbf{p}_t - \mathbf{p}_{t-1}\|_1^2$. Moreover, $\overline{\mathbf{w}}_T$ has a non-negative margin when $T = \Omega(\sqrt{\log n}/\gamma)$ and $\overline{\gamma}(\mathbf{v}_T) = \Omega(\gamma - \frac{8\log n}{\gamma T(T+1)})$.*

---

**Algorithm 3** NAG

---

**Input**: $\mathbf{v}_0 = \mathbf{0}, \mathbf{s}_0 = \mathbf{0}$.
**for** $t = 1, \ldots, T$ **do**
  $\mathbf{u}_t = \mathbf{s}_{t-1} + \frac{1}{2(t-1)}\mathbf{v}_{t-1}$
  $\mathbf{v}_t = \mathbf{v}_{t-1} - \eta_t \nabla R(\mathbf{u}_t)$
  $\mathbf{s}_t = \mathbf{s}_{t-1} + \frac{1}{2(t+1)}\mathbf{v}_t$
**end for**
**Output**: $\mathbf{s}_T$

---

$$\mathsf{OL^P} = \mathsf{OFTRL}\left[E(\cdot), \tfrac{1}{n}, \tfrac{1}{4}, \ell_{t-1}(\cdot), \Delta^n\right] \Leftrightarrow \mathbf{p}_t = \operatorname*{argmin}_{\mathbf{p} \in \Delta^n} \frac{1}{4}\left[\sum_{s=1}^{t-1} \alpha_s \ell_s(\mathbf{p}) + \alpha_t \ell_{t-1}(\mathbf{p})\right]$$
$$+ \mathcal{D}_E\left(\mathbf{p}, \tfrac{1}{n}\right)$$

$$\mathsf{OL^w} = \mathsf{FTRL}^+[0, 0, 1, \mathbb{R}^d] \Leftrightarrow \mathbf{w}_t = \operatorname*{argmin}_{\mathbf{w} \in \mathbb{R}^d} \sum_{j=1}^{t} \alpha_j h_j(\mathbf{w})$$

**Output:** $\widetilde{\mathbf{w}}_T = \frac{\sum_{t=1}^{T} \alpha_t}{4}\overline{\mathbf{w}}_T$.

---

## 4.4 Accelerated Perceptron of Yu et al. (2014)

Finally, Yu et al. (2014) considers applying the mirror-prox algorithm to solve the max-min optimization problem:

$$\max_{\|\mathbf{w}\|_2 \le 1} \min_{\mathbf{p} \in \Delta^n} \mathbf{p}^\top A\mathbf{w}, \tag{8}$$

As discovered by Rakhlin & Sridharan (2013), mirror-prox can be recovered by their optimistic online mirror descent (OMD) with an appropriately chosen optimistic term. Here, we show that, by simply manipulating the notation, the algorithm of Yu et al. (2014) can be recovered as two-players applying OMD to solve (8). Due to page limitation, the algorithmic details and their analysis are postponed to Appendix C.5, which we summarize as follows (formalized in Theorems 8 and 9):

**Theorem 5** (informal). *Let $g(\mathbf{w}, \mathbf{p}) = \mathbf{p}^\top A\mathbf{w}$. The Perceptron of Yu et al. (2014) can be described under Protocol 1, where both players apply OMD endowed with the appropriate parameters. Moreover, under Assumption 1, $\overline{\mathbf{w}}_T$ has a non-negative margin when $T = \Omega\left(\frac{\sqrt{\log n}}{\gamma}\right)$.*

## 5 Beyond Perceptrons

In This section, we show that our framework benefits a wide range of other problems.

### 5.1 Margin Maximization by Nesterov's Accelerated Gradient Descent

One major task for the implicit bias study is to explain why *commonly-used* first-order optimization methods generalize well. For linearly separable data, previous work proves that GD and momentum-based GD prefers the $\ell_2$-maximal margin classifier by showing that they can maximize the margin. In this part, we show that the well-known Nesterov's accelerated gradient descent (NAG), with appropriately chosen parameters, can maximize the margin in an $O(1/\gamma^2)$ rate. Specifically, following previous work, we consider maximize the margin by solving the ERM problem in (7) and apply Nesterov's accelerated gradient descent (NAG) to minimize the objective function. The details of the algorithm are summarized in the first box of Algorithm 3, and its equivalent form under Protocol 1 is presented in the second box of Algorithm 3. For the NAG algorithm, we set the objective function as $g(\mathbf{w}, \mathbf{p}) = \mathbf{p}^\top A\mathbf{w} - \frac{1}{2}\|\mathbf{w}\|_2^2$, and have the following conclusion.

**Theorem 6.** *Let $\eta_t = \frac{t}{R(\mathbf{u}_t)}$ and $\alpha_t = t$. Then, the two expressions in Algorithm 3 are equivalent, in the sense that $\mathbf{s}_T = \widetilde{\mathbf{w}}_T$. Moreover, $\frac{\min_{\mathbf{p} \in \Delta^n} \mathbf{p}^\top A\mathbf{s}_T}{\|\mathbf{s}_T\|_2} \ge \gamma - \frac{8\log n + 2}{T(T+1)\gamma}$.*

---

**Algorithm 4** Accelerated algorithm for the $p$-norm perceptron

$$\mathsf{OL}^{\mathbf{w}} = \mathsf{OFTRL}\left[\frac{1}{2(q-1)}\|\cdot\|_q^2, \mathbf{0}, \eta^{\mathbf{w}}, h_{t-1}(\cdot), \mathbb{R}^d\right] \Leftrightarrow$$

$$\mathbf{w}_t = \underset{\mathbf{w}\in\mathbb{R}^d}{\operatorname{argmin}} \eta^{\mathbf{w}} \sum_{j=1}^{t-1} \alpha_j h_j(\mathbf{w}) + \alpha_t h_{t-1}(\mathbf{w}) + D_{\frac{1}{2(q-1)}\|\cdot\|_q^2}(\mathbf{w}, \mathbf{0})$$

$$\mathsf{OL}^{\mathbf{p}} = \mathsf{FTRL}^+\left[E, \frac{\mathbf{1}}{n}, \eta^p, \Delta^n\right] \Leftrightarrow \mathbf{p}_t = \underset{\mathbf{p}\in\Delta^n}{\operatorname{argmin}} \eta^{\mathbf{p}} \sum_{s=1}^{t} \alpha_s \ell_s(\mathbf{p}) + \mathcal{D}_E\left(\mathbf{p}, \frac{\mathbf{1}}{n}\right)$$

**Output:** $\overline{\mathbf{w}}_T = \frac{1}{T}\sum_{s=1}^{T} \mathbf{w}_s$.

---

**Remark** Theorem 6 indicates that NAG can also maximize the margin in an $O(1/T^2)$ rate. Note that, *in Algorithm 3, the* **p***-player plays first and the* **w***-player second, while, in Algorithm 2, it is the other way around*. This difference makes sense as Algorithm 3 can be considered as applying Nesterov's acceleration in the dual space (Ji et al., 2021), while Algorithm 3 uses Nesterov's acceleration in the *primal space*.

## 5.2 ACCELERATED $p$-NORM PERCEPTRON

In this section, we focus on the $p$-norm Perceptron problem, introduced by Gentile (2000). Compared to the classical perceptron problem, $p$-norm Perceptron introduces the following assumption, which is more general than Assumption 1.

**Assumption 2.** *For the p-norm Perceptron problem, we assume:* $\forall i \in [n]$, $\|\mathbf{x}^{(i)}\|_p \leq 1$, *where* $p \in [2, \infty)$. *Moreover, assume there exists a* $\mathbf{w}^* \in \mathbb{R}^d$, *such that* $\|\mathbf{w}^*\|_q \leq 1$ *and* $\min_{\mathbf{p}\in\Delta^n} \mathbf{p}^\top A\mathbf{w}^* \geq \gamma$. *Here,* $\|\cdot\|_q$ *is the dual norm of* $\|\cdot\|_p$; *i.e.,* $\frac{1}{p} + \frac{1}{q} = 1$ *and* $q \in (1, 2]$.

Under Assumption 2, Gentile (2000) proposes a mirror-descent style algorithm that achieves an $\Omega((1-p)/\gamma^2)$ convergence rate. In the following, we provide a new algorithm with a better rate. Specifically, under Protocol 1, we define the objective function as $g(\mathbf{w}, \mathbf{p}) = \mathbf{p}^\top A\mathbf{w}$, and introduce Algorithm 4, wherein the **w**-player uses OFTRL with regularizer $\frac{1}{2(q-1)}\|\cdot\|_q^2$, which is 1-strongly convex w.r.t. the $q$-norm (Orabona, 2019). On the other hand, the **q**-player employs the FTRL$^+$ algorithm. We have the following result.

**Theorem 7.** *Let* $\alpha_t = 1$, $\eta^{\mathbf{p}} = 1/\eta^{\mathbf{w}}$, *and* $\eta^{\mathbf{w}} = \sqrt{\frac{1}{2(q-1)\log n}}$. *Then the output* $\overline{\mathbf{w}}_T$ *of Algorithm 4 has a non-negative margin when* $T = \Omega\left(\sqrt{2(p-1)\log n}/\gamma\right)$.

## 6 CONCLUSION

In this paper, we provide a unified analysis for the existing accelerated Perceptrons, and obtain improved results for a series of problems. In the future, we will explore how to extend our framework to other closely related areas, such as semi-definite programming (Garber & Hazan, 2011) and generalized margin maximization (Sun et al., 2022). Another interesting direction is to consider whether other more advanced online learning algorithms (Orabona & Pál, 2016; Cutkosky & Orabona, 2018; Jun et al., 2017; Zhang et al., 2018; Wang et al., 2020; Zhao et al., 2020) is useful for perceptron and the implicit bias analysis based on our framework.

### ACKNOWLEDGMENTS

We gratefully thank the AI4OPT Institute for funding, as part of NSF Award 2112533. We also acknowledge the NSF for their support through Award IIS-1910077, as well as an ARC-ACO fellowship provided by Georgia Tech.

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
