# OpenReview forum: "On Accelerated Perceptrons and Beyond"
_ICLR.cc/2023/Conference — ICLR 2023 poster_

### Official Review · Reviewer_nHQ8 · 2022-10-18

**Confidence:** 4
**Correctness:** 4
**Technical Novelty And Significance:** 4
**Empirical Novelty And Significance:** Not applicable
**Recommendation:** 8

**Clarity, Quality, Novelty And Reproducibility:**

***Clarity***
The writing is overall clear enough. Some minor flaws:
1. Using $\Omega$ to write upper bounds of the iteration complexity is unusual. I think most of the $\Omega$'s should be replaced by $O$'s.
2. Second paragraph on p. 3: The $O ( \mathrm{poly} (n) \log ( 1 / \gamma ) )$ rate seems to be wrong, as it does not vanish.
3. I suggest writing $w$ and $p$, instead of $v$ and $q$, in (4) to keep the notations consistent.
4. First line in Algorithm 2: The $\theta_t$ and $\beta_t$ should not be inputs. They are computed in the iterations.
5. p. 9: I do not understand the sentence "we consider the ERM problem in (7)..." I suppose the authors are not solving (7) in Section 5.1.
5. I think it is better to explicitly state that the function $g$ are the same in Section 4.2, Section 4.3, and Section 5.1.

***Quality***
The paper is well written.

***Novelty***
The paper is novel enough to me. One may be concerned about the technical novelty. The no-regret learning interpretations are reminiscent of the works by Abernethy and Wang; in particular, the unusual FTRL+ algorithm, unrealizable in standard online convex optimization problems, was also considered by Abernethy and Wang. Nevertheless, this paper does not directly apply existing results; the technical details are different enough and the unified interpretation of the algorithms by by Soheili & Pena (2012) and Ji et al. (2021) was not obvious to me.

***Reproducibility***
This is a theory paper.

**Strength And Weaknesses:**

***Strength***
The problem is classic. The results are new and interesting. The unified interpretation provides new insights.

***Weaknesses***
I don't see any significant weakness. Due to the *unreasonably tight review deadline*, I only checked the proof of Theorem 3.

**Summary Of The Paper:**

Consider the problem of finding a linear classifier with positive margin for a given linearly separable dataset, which can be formulated as a bilinear minmax problem. This paper proposes a unified interpretation for two algorithms by Soheili & Pena (2012) and Ji et al. (2021) and a new algorithm based on Nesterov's accelerated gradient method. The key idea is to consider a slightly modified minmax problem---a game---and solve it via no-regret learning. The idea is reminiscent of the papers by Abernethy and Wang on interpreting convex optimization algorithms via no-regret learning in Fenchel games.

**Summary Of The Review:**

The paper considers a classic problem, is novel, and provides new insights for understanding existing algorithms. I don't see any significant weakness but only some minor flaws in the presentation. Therefore, I suggest acceptance of this paper.

---

> ### Author Response · Authors · 2022-11-16
> **Response to Reviewer nHQ8**
>
> Thank you for the constructive review and supportive comments about our work! We will revise the paper accordingly.
>
> For the problems on clarity:
> 1. Thank you for the suggestion. We use $\Omega$ to show that the algorithm can find a perfect classifier if $T$ is larger than a certain value. But $O$ is more common and we will switch to this notation.
> 2. Thank you for the catch. With our notation, it should be $\Omega(poly(n)\log(1/\gamma))$. $\gamma$ is usually very small (smaller than 1), so $\log(1/\gamma)$ is better than $1/\gamma^2$. We will add more discussion to make it clearer.
> 3. Thank you for the suggestion. We will switch the notation.
> 4. Thank you for the suggestion. We will revise the algorithm description.
> 5. Yes, you are right, the algorithm is to minimize (7) with NAG, but the real goal is to examine the margin maximization rate. We will make it more clear.
> 6. Thank you and we will add the exact definition of $g$ in Section 4.2, Section 4.3, and Section 5.1.

---

### Official Review · Reviewer_gCQu · 2022-10-22

**Confidence:** 4
**Correctness:** 4
**Technical Novelty And Significance:** 4
**Empirical Novelty And Significance:** Not applicable
**Recommendation:** 8

**Clarity, Quality, Novelty And Reproducibility:**

The paper was well-written and easy to follow throughout. The approach is novel and unifies a number of existing works using relatively simple analysis.

**Strength And Weaknesses:**

The paper is very strong overall. The writing is clear and the math is easy to follow (particularly for those already familiar with optimistic MD/FTRL). The analysis is elegant and leads to novel results while capturing existing works as special cases.

My only issue with the paper is that there are some connections to recent works on acceleration that I think would be worth discussing in the paper. In particular, Allen-Zhu (2016) and Joulani (2020) similarly view accelerated algorithms as decomposing into two online learning algorithms with interacting dynamics, with optimistic FTRL being the key component in Joulani 2020. I think these are both similar in spirit to what's happening here and would be worth discussing (or at least citing).

- Joulani, P., Raj, A., Gyorgy, A., & Szepesvari, C. (2020). A simpler approach
  to accelerated optimization: iterative averaging meets optimism. In H. D. III,
  & A. Singh, Proceedings of the 37th International Conference on Machine
  Learning (pp. 4984–4993). : PMLR.
- Allen-Zhu, Z., & Orecchia, L. (2016). Linear coupling: an ultimate unification
  of gradient and mirror descent.


**Summary Of The Paper:**

This paper introduces a unified analysis for accelerated perceptrons via reduction to a certain minmax game. Improved rates are then achieved by designing the dynamics of the min and max player. The analysis is simple and easily captures several prior works as special cases, while also easily enabling the authors to develop new algorithms with improved rates.

**Summary Of The Review:**

This is an excellent paper and I highly recommend its acceptance.

---

> ### Author Response · Authors · 2022-11-16
> **Response to Reviewer gCQu**
>
> Thank you for the constructive review and supportive comments about our work! We will revise the related work section and add the citations and discussion as mentioned.

---

### Official Review · Reviewer_PKdM · 2022-11-10

**Confidence:** 2
**Correctness:** 3
**Technical Novelty And Significance:** 2
**Empirical Novelty And Significance:** 2
**Recommendation:** 5

**Clarity, Quality, Novelty And Reproducibility:**

I think the paper tries to cover multiple earlier works using minimax perspective (although some of the algorithms are not necessarily directly solving the minimax problem). I find it difficult to understand its consequence in optimization.

First it does not explain well what's especial for this problem that makes its analysis beyond existing analysis for minimax problem.

Second some online terminologies are used which is different from what would have been used in the optimization literature, and makes it somewhat more difficult for researchers in optimization to make connection to this paper. I am not sure this is necessary as the authors did not treat the problem as an online learning problem at all, but only use online learning as a technical tool to prove optimization convergence results.

Third many different results that are not coherent are discussed. I am not sure that's necessary as it looses focus.

What I have missed is that what new idea does this work bring to minimax optimization?

**Strength And Weaknesses:**

Strength. The minimax formulation is an interesting view of what perceptron tries to optimize, although it is fairly straight-forward.

Weakness
I am not sure what's the main message. Minimax problem is well-studied in optimization. This particular problem is a very specialized formulation of minimax problem. It will be nice to understand what makes it special and why it is not trivial to use some results for general minimax problems to derive convergence for this problem. Are there really new algorithms or merely reinterpretation of standard results for this specialized problem? If the latter, what's the benefit of the specialization?

**Summary Of The Paper:**

The paper establishes minimax object function for perceptron algorithm, and treat it as a minimax optimization problem. For this problem, faster convergence rate (than perceptron) can be obtained.

**Summary Of The Review:**

I think the paper has potential, and tries to make interesting interpretation and connection to minimax optimization of a very specialized problem related to the perceptron algorithm.

However, due to the content of the paper, it should be regarded as an optimization paper. For such a paper, the problem considered is quite narrow, so I'd expect more, including  what's special about this problem beyond standard minimax optimization, and what are the new optimization methods this paper actually develops in the broader context of minimax optimization. As it is currently written, I find it hard to interpret its significance.

---

> ### Author Response · Authors · 2022-11-16
> **Response to Reviewer PKdM (part 1)**
>
> >“Summary Of The Paper: The paper establishes minimax object function for perceptron algorithm, and treat it as a minimax optimization problem. For this problem, faster convergence rate (than perceptron) can be obtained.” “why it is not trivial to use some results for general minimax problems to derive convergence for this problem”
>
> Thanks for the summary. Faster convergence rate (than perceptron) has already been established by previous work, so it is not a contribution of this paper. Moreover, merely re-deriving the convergence for Perceptron under the min-max optimization framework is also not our goal. Therefore, we believe that the core contributions of our paper are missed in the current review, so we would like to emphasize them as follows.
>
> Existing accelerated Perceptrons are developed under different objective functions with different optimization techniques. In this paper, we construct a unified game framework under which all existing accelerated Perceptrons can find their **exact equivalent forms**. After obtaining the equivalent forms, deriving the convergence rate is indeed straightforward. However, proving such an equivalence, and finding the correct online optimization algorithms, are challenging (i.e., it is highly non-trivial to show these different algorithms can indeed be perfectly described under the no-regret game framework). We believe the unified view itself is already novel and brings lots of new insights. Moreover, it also has the following *additional merits*:
> * **Simplifying the analysis**: Under the proposed framework, the proof of existing accelerated perceptron algorithms can be significantly simplified, as the convergence can now be directly obtained via plugging in the regret bounds of the well-studied online learning algorithms;
> * **Drawing connections between algorithms**: We show that two existing accelerated Perceptrons are essentially the same, which, as confirmed by Reviewer nHQ8, is only obvious under our framework;
> * **Leading to improved results for p-norm Perceptron**: The framework leads to a new algorithm for the p-norm Perceptron problem, and we show it has an improved convergence rate compared to the existing methods (from $O(1/\gamma^2)$ to $O(1/\gamma)$).
> * **Leading to two novel results for the implicit bias analysis**: Since our framework can interpret the algorithm of Ji et al. (2021), we naturally find it is also helpful for implicit bias analysis. We contribute to this problem in the following ways: a) We show that, under our analysis framework, we can easily improve the margin maximization rate of the algorithm of Ji et al. (2021) from $O(\log t/t^2)$ to $O(1/t^2)$; b) Through our framework, we show that Nesterov’s accelerated gradient descent (NAG) also enjoys an $O(1/t^2)$ margin maximization rate. To our knowledge, it is the first time the implicit bias property of NAG is proved. c) Finally, we note that the connection between implicit bias and game is also new. The game theoretic analysis by nature facilitates the implicit bias analysis since it allows for a more unified treatment of the implicit bias of the algorithms that can be described using the game dynamics.
>
> We hope the above summary could answer the questions/concerns raised in the review. Next, we also would like to response to the following comments:
>
> > "Are there really new algorithms or merely reinterpretation of standard results for this specialized problem? If the latter, what's the benefit of the specialization?"
>
> Please see Response 1 for the new algorithms and improved results we provide. Moreover, we believe novelty is not only about providing SOTA rates or brand-new algorithms (although we do provide several new/accelerated results): Drawing connections between different algorithms/problems, providing a unified/universal view for analyzing existing methods, are also novel and will motivate the researchers in related areas.
>
> > "many different results that are not coherent are discussed. I am not sure that's necessary as it looses focus."
>
> Thank you for the comments. To summarize, in this paper, we consider three problems: Perceptron, implicit bias, and the p-norm Perceptron. The three problems are strongly connected, as they all stem from the linear separation setting, but their goals and assumptions are different. Perceptron tries to find a zero-error classifier; implicit bias focuses on the margin maximization rate; p-norm Perceptron is a variant of Perceptron with different assumptions on the optimal classifier. In this paper, we reveal that all three problems can be solved under one framework,  and existing algorithms for these problems can be unified. We start from Perceptron, and then extend the framework to the other two settings. Therefore, we believe our presentation is coherent, and the many results show the powerfulness and potential of the proposed framework.

---

> > ### Author Response · Authors · 2022-11-16
> > **Response to Reviewer PKdM (part 2)**
> >
> > > “What I have missed is that what new idea does this work bring to minimax optimization?” “However, due to the content of the paper, it should be regarded as an optimization paper. For such a paper, the problem considered is quite narrow, so I'd expect more, including what's special about this problem beyond standard minimax optimization, and what are the new optimization methods this paper actually develops in the broader context of minimax optimization.”
> >
> > This paper focuses on several classical and important ML problems, instead of the general min-max optimization. We use the game as a tool to unify many different algorithms under the same framework. The connection between these algorithms is new, and it is challenging to prove that these algorithms can be exactly written as two players applying online learning algorithms to solve games. Apart from a unified view/drawing connection among different algorithms, the framework also simplifies the analysis, and leads to new results for several other related problems, including implicit analysis and p-norm Perceptron.

---

### Decision · Program_Chairs · 2023-01-20

**Decision:**

Accept: poster

**Justification For Why Not Higher Score:**

Technical novelty not high enough.

**Justification For Why Not Lower Score:**

Always hard to find a significant contribution on a field that has been long studied.

**Metareview: Summary, Strengths And Weaknesses:**

This paper proposes a formulation of the Perception algorithm as a minimax optimization problem, that is solved thanks to the use of Nesterov's accelerated method.

+ Novel contribution in a field that has been studied for long
+ Paper clearly written
- Empirical test could have been performed.

**Note From Pc:**

if the above contains the word "oral" or "spotlight" please see: "oral" presentation means -> notable-top-5% and "spotlight" means -> notable-top-25%. As stated in our emails, we are disassociating presentation type from AC recommendations